# Analytical and Native Concepts in Argentina's Post-Conciliar Catholicism: The Case of "Liberationism", "Popular Pastoral Theology", and "Theology of the People"

Claudio Iván Remeseira 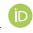

School of Social Sciences, University of Buenos Aires, Buenos Aires CABA 1075, Argentina; remeseira@gmail.com

**Abstract:** "Liberationism", a term derived from Liberation theology (LT), is an analytical concept used by religious historians and sociologists as a generic designation for Latin American post-conciliar Catholicism. "Theology of the People" (TP) designates a theological school created in Argentina during the late 1960s by the Episcopal Pastoral Commission (COEPAL), although the term used by its members was not TP but "Popular Pastoral theology" (TPp). Successive generations of theologians developed new versions of TPp ("popular piety theology", "theology of culture", etc.). I call those versions the diachronic variants of TP, and I regard TP as their synchronic representation. TP has been called the "Argentine School of Liberation Theology", but there are substantial differences between TP and TL. In this paper, I argue that it is inaccurate to use the term "liberationism" to refer to TP because that term alludes to LT's model of inter-relations between religion and social change, a model explicitly rejected by the creators of TP. I frame the theoretical discussion on the use of analytical and native concepts in Quentin Skinner's linguistic contextualism perspective and I explain the differences between TP and TL in the context of the theological–political debates in late-1960s Argentina around the issue of popular Catholicism.

**Keywords:** liberationism; liberation theology; theology of the people; popular pastoral theology; post-conciliar Catholicism; Latin American theology; analytical and native concepts; COEPAL; ECOISYR

## 1. Introduction

Liberationist Christianity, or liberationism, has been defined by Löwy as "the social movement that has its intellectual expression in liberation theology [hereafter, LT] and criticizes the 'really existing' modernity in Latin America (dependent capitalism) in the name of pre-modern values and the promise of a utopian modernity (the classless society) through the socio-analytical mediation of Marxist theory, which unites the critique of the first with the promise of the second" (Löwy 1999, p. 64). This definition has been widely adopted by historians and religion sociologists to designate the different brands of the "progressive" theology that emerged in Latin America during the first long decade of the post-conciliar period (roughly, from 1965 to 1979), including the theology of the people [hereafter, TP]. The goal of this article is not to discuss to what extent this definition accurately describes the different expressions of LT over the years[1], but to demonstrate that applying the analytical term "liberationism" to TP is inaccurate and misleading because it ignores crucial differences between LT—as defined by Löwy—and TP, therefore obscuring the latter's meaning.

TP was originally developed by the team of theological experts or *periti* of the National Pastoral Commission (COEPAL, for its acronym in Spanish), created in 1967 by the Argentine Episcopal Conference to implement in that country the resolutions of Vatican II.[2] The name that those experts gave to their theological reflections was not TP but popular pastoral theology (TPp) (Boasso 1974). In the following decades, this school of theology was also known by other names, such as), the Argentine School of popular pastoral (Alliende

1976), populist theology (Oliveros 1977), theology from the praxis of the Latin American peoples (Scannone 1976, [1982] 1983), theology of culture, and popular religiosity theology (Scannone 1974a, 1990a). By the turn of the century, and especially after the election to the papacy of Jorge Mario Bergoglio, the term "theology of the people" became the overarching denomination for all the developments of this theological school since its inception (Scannone 2013, 2014a, 2014b, 2015a, 2015b, 2015c, 2015d, 2017, 2019; Zanca 2022). Here, I will use "TP" with a *synchronic* perspective, to refer to all these developments considered as a whole and coherent body of theological thought, and I will use "TPp" or any other variety of TP under a *diachronic* perspective, focused on a specific version of that theological thought, anchored in a specific historical period.[3]

Methodologically, my rejection of the use of "liberationism" to refer to TP in any of its historical variants is based on a contextualist perspective, inspired by the Cambridge School of intellectual history. According to the main representative of this historiographical current, "any text is always primarily *an intervention in an argument*, and the most interesting question is always to ask about the character of the intervention." (Skinner 1997, pp. 71–72; my italics). For Skinner, the historiographical task basically consists in restoring a certain text to its original context and in determining what the author was trying to *do* with that text, i.e., reconstructing the meaning of the text and its author's beliefs and intentions (Skinner 2006, p. 125).

Any relevant reconstruction of the linguistic context requires in turn to distinguish between analytical and native concepts. The relevance of this distinction in the Argentine historiography of this period has already been noticed with regard to other native concepts, such as national populism (Amaral 2018), the Peronization of university students (Friedemann 2017b), the vernacular New left (Martínez 2020), and "socialismo national"[4] (Caruso 2022). The present article shares the theoretical concerns of those works.

I will divide my demonstration into two parts. First, I will address the relationship between LT and TP and the debate over Marxism and Marxist theory in their early days (the late 1960s–early 1970s). This was a theological–political and methodological dispute that shook the foundations of the Argentine and Latin American Church and had long-term implications for LT and TP standing within the Church's official teaching. In the second part, I will describe the debate over the nature of popular Catholicism between the COEPAL's *periti* and Aldo Büntig, one of the earliest proponents of LT. The *periti* rejected Büntig's secularist outlook of religious phenomena and emphasized instead the supernatural character of popular faith and the importance of popular culture as a historical carrier of those supernatural values (Gil and Bender 1999).[5] My purpose in this paper is not to survey and determine the peculiarities of the *periti*'s conception of popular Catholicism of popular culture vis a vis those of other LT theologians, but to delve into a contextual analysis of their debate with Büntig in order to expose the mainstays of their theological thought. In other words, I will focus on a diachronic analysis of TP and LT. Further developments of LT, less closely connected with the Marxist theory (Segundo 1984; De Schrijver 1998), as well as the so-called convergence between TP and LT under Francis' pontificate (Løland 2021) are of course relevant from a synchronic perspective but do not invalidate the diachronic analysis of their differences—and similarities—within the linguistic context of their historical emergence.

## 2. The Methodological and Theological–Political Debate

### 2.1. Is TP a Variant of LT or an Autonomous Theological School?

TP, under the different names by which it has been known since its creation, has been described as a current of LT (Oliveros 1977; Lehmann 1978; Scannone [1982] 1983, 1987a, 1987b; Gutiérrez 1988), and more specifically, as the "national and popular line" of said theology (Methol Ferré 1982). Others, both defenders (Politi 1992; Scannone 2017, 2019) and critics of TP—including the creator of the term "theology of the people", Uruguayan Jesuit Juan Luis Segundo (Segundo 1975)—emphasize instead the differences between the two. Historically, the latter perspective proved to be more accurate: in the heat of the theological–



political debates of the 1970s and 1980s of the past century, LT and TP would be seen within the Church as two antagonistic versions of Latin American post-conciliar theology.

In the beginning, this was not the case. The theologians who in the wake of Vatican II began to develop what would later be known as TL and TP, drew on a common pool of ideas ascribed to the so-called "progressive Catholicism" of that time, such as dependency theory, anti-imperialism, anti-colonialism and the ubiquitous concept of "liberation", a concept that would remain at the center of the theological debate for the next couple of decades.[6]

The term "liberation theology" gain traction internationally after the publication of Gustavo Gutiérrez's book by that title (Gutiérrez [1971] 2015). Shortly before the appearance of Gutiérrez's book, the same title had been used in a lecture by Argentine bishop Eduardo Pironio (Pironio 1970), then secretary general of the Latin American Episcopal Council (CELAM, its acronym in Spanish). The fact that Pironio had a key role in the dissemination of TP in Latin America—and, even more importantly, in its incorporation (through the notion of popular religiosity)[7] to papal magisterium[8]—suggests that during the early years of the post-conciliar period, Latin American theologians still shared a common frame of reference. In any case, over the 1970s LT became a metonym for Latin American theology, and later for progressive theology, especially, but not exclusively, in the Third World.

LT has four defining characteristics: (1) the claim of a preferential option for the poor; (2) an emphasis on the social consequences of Scripture; (3) the idea that praxis or action precedes theological reflection, and (4) the prophetic commitment, i.e., the denunciation of the structures of oppression, originally understood as socio-economic oppression but later extended to all kinds of oppression: racial, gender, etc. (Egan 2009, pp. 76–77). The coexistence under the same denomination of such a broad spectrum of ideas has led some to speak of *theologies* of liberation (Scannone [1982] 1983, p. 276), but the term continues to be used, both within and outside the specialized field, in the singular. Complicating the matter even further, many authors have "evolved" over the years in regard to how they define this term.[9]

The term "theology of the people", with its original-dismissive bent, was vindicated two decades later by Sebastián Politi, a disciple of one of COEPAL's leading *periti*, Lucio Gera (Politi 1992). Politi's book, based on his graduate dissertation for the Buenos Aires Metropolitan Seminary, was prologued by Gera, who had also supervised his dissertation.[10] In the prologue, Gera establishes a direct link between what Politi calls the theology of the people and the popular pastoral theology (although he does not call it by that name) produced by the COEPAL two decades earlier.[11] The term, initially limited to Argentine theological circles, spread globally after the election of Jorge Mario Bergoglio to the papacy.

Generally speaking, the four characteristics highlighted above can be applied to TP and all of its diachronic variants (TPp, popular piety theology, theology of culture, etc.); in this sense, it is legitimate to claim that TP is a variant of LT. However, when we move from the generalities to the specificity of the arguments, the differences between one and the other emerge in a clear and unmistakable way.

The key difference lies in the third postulate. By the 1960s, philosophers and *nouvelle theologie* practitioners had already argued that existence and action were *the* constitutive dimensions of human life; the first generation of liberation theologians built up on that legacy and added the conceptual apparatus of Marxism, considered then by many as "the [insurmountable] formal framework of all philosophical thought" (Gutiérrez [1971] 2015, p. 8).[12] This formal framework included the concepts of social class and class struggle. However, TP will be developed *against* these concepts, which it will respectively counter with a non-Marxist notion of *people* and with the idea of the unity of *nation* over its social class divisions, a recast of the old organicist theses of Catholic corporatism and the "myth of the Catholic nation" (Zanatta 1999).

COEPAL's theologians regarded the Marxist perspective of the nascent LT as "sociologism" and dismissed it as a mix of abstract, universal ideas and conceptual schemes, which they considered foreign to Argentina's (and by extension, Latin American) history

and cultural traditions—contrary, in a word, to its *Volksgeist*.[13] To the "sociologism" of that "imported" (as opposed to home-grown) ideas, they opposed their "historical-cultural method", a recall of the ideographic role of history as a paradigm for social sciences (Remeseira 2020).[14]

Through these arguments, the COEPAL's *periti* and their allies in the clergy and the academy were intervening in two major discussions of the early 1970s. The first was a theological–political debate raging within the Third World Priests' Movement (MSTM), the leading left-wing clerical movement in Argentina to which Gera, O'Farrell, Tello, and other important members of the COEPAL were connected (Pontoriero 1991; Martín 1992, 2013; Touris 2021). That debate veered mostly around whether Juan D. Perón, a former President who had lived in exile since his overthrow in 1955 by a civic and military uprising but remained the most popular figure for the Argentine working class, was ideologically able to lead the country through a socialist revolutionary process, at a time when the paradigm for such a process was the Cuban Revolution and, tragically briefly, the Chilean "democratic via to Socialism". The second was a debate on the role that social sciences in general and Marxist theory in particular should have in the formulation of LT. Although *prima facie* methodological, this scholarly debate would eventually provide the conceptual framework for the larger theological–political controversy.

### 2.2. Marxism, Peronism, and the Political–Theological Debate

The first debate, known as the Peronism–Socialism debate (Bresci 1994, pp. 163–67), pitched a vocal minority of left-wing members of the MSTM against its Peronist majority (Pontoriero 1991, p. 97). The "Third World", as the MSTM was colloquially known, was a loose "constellation" of priests (Touris 2021) who advocated for a radical implementation of Vatican II and stood against the more powerful conservative wing of the Argentine Church.[15] The Peronism–Socialism debate was therefore a controversy among self-defined progressives.[16]

The MSTM was born during the dictatorship known as "Revolución Argentina" (1966–1972). Opposition to the military rulers kept the "Third World" factions together and overpowered their inner disputes, but the end of the dictatorship accelerated their centrifugal trends. Perón returned to the country in November 1972, and on March 11 of the following year, a Peronist-led front won the national elections, bringing Héctor J. Cámpora, a Peron's appointee, to the presidency. Despite the return of democracy, the guerrilla organizations that had fought against the military dictatorship—most prominent of all, the Peronist-leftists Montoneros—refused to give up the armed struggle and urged for a deeper radicalization of the new government. Amid growing instability, President Cámpora, considered too lenient with Montoneros and the Peronist left-wing, was forced to resign, and new elections were called for September. Yet not even the election of Perón for an unprecedented third term as President was able to stop the escalation of political violence. In this rarified environment, the tensions between the leftist and Peronist factions of the MSTM grew even more bitter. The Peronism–Socialism debate came to an abrupt end at the VI National Encounter of the MSTM, held at Santa Fe province in August 1974 when the Peronist group walked out of the sessions.

From a theoretical perspective, however, the most important aspect of the Peronism–Socialism debate did not take place at the MSTM meetings but in the academy, especially at the Cátedras Nacionales de Sociología (literally, the National Chairs of Sociology, hereafter CN), a series of courses and seminars taught by a number of Peronist social scientists at the University of Buenos Aires School of Sociology between the mid-1960s and the early 1970s (O'Farrell 1970; Argumedo 1971; O'Farrell et al. 1972; González 1997; Moscona 2010; Friedemann 2017a; Ghilini and García 2008; Ghilini 2011, 2015, 2021; Gómez 2016). The founder of the Cátedras Nacionales was Justino O'Farrell, who alongside Gera and Tello was one of the major "poles" of COEPAL's team of periti (González [2005] 2010, pp. 75–83). One of the CN faculty members laid out the political rationale for the dispute: if Marxism is both the most advanced theoretical framework for social analysis and the synthesis of

revolutionary praxis, how could Marxist intellectuals and activists reject Peronism, the quintessential expression of Argentina's popular struggle for Liberation? For the author, this blatant contradiction between the *diktat* of a revolutionary elite and the majoritarian political choice of the Argentine people undermined not only the explanatory value of Marxism but also its claims to scientific status and universality (Argumedo 1971).

A similar debate erupted among theologians. By the early 1970s, a growing number of new books and scholarly articles had created a critical mass of what was quickly known as LT (Alves 1969, 1970; Assmann 1971a, 1971b, 1972a, 1972b, 1973a, 1973b; Boff 1972, 1975; Büntig 1970, 1973; Comblin 1970, 1973; Dussel 1972, 1973a, 1973b, 1974; Galilea 1973a, 1973b; Gutiérrez 1971, 1973, Gutiérrez [1971] 2015; Segundo 1972, 1973a, 1973b, 1973c, 1975).[17] This body of work constitutes the larger argument into which the COEPAL's *periti* intervened with the creation of TPp. For the analysis of Latin American social reality, most of those works relied on sociological analysis heavily indebted to Marxist theory; in some cases, they overly conflated Marxism and Christian beliefs and ideas.[18] Disagreements over this point emerged early on at international gatherings of theologians, such as the momentous 1972 El Escorial Encounter in Spain (Ames Cobián et al. 1973; Borrat 1973a, 1973c; Scannone 1973c) and in Catholic publications, such as the Uruguayan *Vispera* (Borrat 1973b; Dussel 1974; Galilea 1973b; Scannone 1973b), whose editorial line was very close to the theological–political views of the COEPAL's theologians.[19]

But if faith is a "liberating praxis of transformation" of the socio-economic a political environment, how can the theological discourse "descend" to the strategic and tactical levels of such a practice without being engulfed in ideologically sectarian positions? Moreover, if we try to keep the theological and the political poles of this equation in a productive tension, how can we avoid falling back into a language that is both "theologically impoverished and socio-analytical and politically useless?" (Borrat 1973b, p. 34, my translation). The starkest alternative to Gutiérrez's vision was displayed by Scannone (Scannone 1973a). His presentation, originally appeared as an article in *Víspera* (Scannone 1973d), was a response to a previous debate he had had with Assmann in 1971, during the II Jornadas Académicas organized by the Schools of Philosophy and Theology of the Jesuit Universidad del Salvador in San Miguel, Argentina (González and Maddonni 2018). The debate, in which other El Escorial participants also intervened (Assmann 1972b; Borrat 1972; Dussel 1972; Scannone 1972a, 1972b) veered around the value of Marxism for the analysis of social reality; this was arguably the first public instance of such a debate among "liberationist" theologians.[20]

Instead of Assmann's (and Gutiérrez's) focus on the notions of class and class struggle, Scannone emphasized the role of popular nationalism in the fight against imperialism. This was of course the same alternative that divided the Peronist and socialist factions of the MSTM. The point is furthered by Comblin, who unknowingly offers an even more clearly articulated exposition of COEPAL's *periti* ideas (Comblin 1973, pp. 101–16). Since the Mexican Revolution, he said, populism, i.e., the wide-ranging ideological vindications of a people in need of redemption, has been the soul of Latin American revolutions. "Latin American 'socialism' is above all anti-imperialism and nationalism" (Borrat 1972, p. 34)

Comblin's presentation, argued Álvarez Bolado, shed light on the meaning of LT'S "family of kindred options": on the one hand, we have those options that, in the manner of Gutiérrez and other like-minded theologians, focus on the "dependency-oppression" living conditions of the poor; on the other, those who, similar to Scannone and the COEPAL's theologians, emphasize the "more spontaneous, popular and utopian" significance of national liberation, a perspective that is "less reducible to a socio-analytical interpretation that seeks to explain a complex historical process in terms of a single ideological or political orientation"; in other words—that rejects a Marxist interpretation (Ames Cobián, *cit*: 17). Unlike Scannone or the COEPAL's *periti*, however, Comblin does not endorse this second option, but he affirms that "only through the lens of Latin American nationalism we can properly understand the present movements and ideologies (...) and how they interact with (...) Marxist and explicitly Christian movements" (Comblin 1973, pp. 101–2, my translation).[21]

Populism is intimately linked to popular religiosity; having experienced Peronism, "the most lasting populist experience ever registered in Latin America" (Borrat cit: idem), Argentine theologians were particularly interested in this issue. The Argentine panelist who spoke on popular Catholicism presented a competing view to the anti-secularist COEPAL perspective (Büntig 1973). We will develop this point in the last section of this paper.

## 3. ECOISYR, COEPAL, and the Debate on Popular Catholicism

### 3.1. Popular Religiosity and Secularization Paradigm

The first clear articulation of the difference between what would later be called LT and TP arose in the debates over the nature of popular Catholicism that took place between Argentine religious sociologists and theologians in the late 1960s. The historical setting of those debates was COEPAL and its predecessor, the Coordinating Team for the Research of Society and Religion (Equipo Coordinador de Investigación de Sociedad y Religión, ECOISYR). The conflicting thesis was originally argued for by two members of the latter group, priests and sociologists Aldo Büntig (Büntig 1968a, 1968b, 1969a, 1969b, 1970; Soneira 1996) and Justino O'Farrell (Gil and Bender 1999). Büntig's overall position can be considered the cornerstone of the modern Argentine sociology of religion.[22] O'Farrell's thesis (Severini 2000), which was later elaborated from a theological standpoint by his COEPAL colleagues Gera and Rafael Tello (Gil and Bender 1999; Forcat 2016), constitutes the foundation of the TPp.[23] The clash of these two positions marks the theoretical dividing line between the Marxist-leaning and Peronist "Third World" priests mentioned above, in this case, not from the methodological perspective but from the perspective of popular faith. [24] In the broader Latin American context, this was the fall-out of the debates at the II CELAM Conference, held in 1968 in Medellín, Colombia[25], which anticipated the lines of force of the popular religiosity debate that would shake the Catholic Church over the next couple of decades.[26]

The post-conciliar context of this controversy in Latin America has been studied in-depth by theologians and historians, but there is another geographically–conceptually broader context that we should also be paying attention to: the debate between sociologists of religion in Europe and the United States that would give rise to the "paradigm of secularization" and the birth of the sociology of religion as a church-independent, scientific discipline. (Tschannen 1992a, 1992b). The intersection of these two contexts brings greater depth to our analysis.[27]

To summarize, for the COEPAL's *periti*, their opponents were "sociologists", incapable of recognizing the supernatural character of the religious faith of the poorest segments of Latin American society; for Büntig and other liberation theologians, popular religiosity was at best a kind of Catholicism corrupted by superstition and non-Christian beliefs, and at worst a pious mask for a conservative social ideology. The wounds opened by this division will only begin to heal after the end of the Cold War. Against this backdrop, the unstoppable advance of secularization and urbanization was undermining a sociological discipline that had been largely monopolized by the different Christian churches.

### 3.2. The ECOISYR and the Sociology of Religion: Aldo Büntig

ECOISYR had been created at the end of 1966 by a group of sociologist priests to provide their professional expertise to the Argentine Catholic Church hierarchy in order to contribute to the implementation of Vatican II resolutions.[28] By the mid-1960s, religious sociology had a small but active set of practitioners in Argentina, trained mainly in Belgium and France (Zanca 2006, pp. 183–89). The use of surveys and other quantitative methods of research had had a long tradition in the Argentine Catholic Church[29], those antecedents served as fertile ground for the growth of the discipline since at least the 1940s.[30] By the late 1950s and early 1960s, these seeds had borne fruit (Amato 1957, 1963, 1965; Donini 1961, 1968; O'Farrell and Antonio 1963).[31] At the same time, and thanks in part to foreign aid, religious sociology had spread throughout Latin America.[32]

ECOISYR had a brief life. In 1967, the Episcopate created the aforementioned COEPAL, which absorbed the former's functions and even some of its members, such as O'Farrell, Alberto Sily, and COEPAL's future Executive Secretary, Gerardo Farrell. However, some of the projects started by ECOISYR were completed after the group disbanded. The most important of these projects was a series of *Cuadernos* (literally, notebooks) on popular Catholicism coordinated by Büntig. The series approached the phenomenon from an interdisciplinary perspective: sociological, biblical, psychological, anthropological, and historical.[33] Büntig also wrote the first of the *Cuadernos*, which includes the theoretical foundation of the entire series (Büntig 1969a).[34]

Popular religiosity had been one of the central themes discussed at Medellín. The conference's final document held an ambiguous position in this regard: it acknowledged the importance of popular Catholicism but called for its "purification" of irrational or semi-pagan beliefs and, above all, weighed upon its potential to hinder social change. The latter is apparent in the division of the document into two parts, dedicated to the Church's social action ("human promotion") and evangelization mission ("promotion of the faith") respectively; this division suggests that social and apostolic work represent two separate dimensions of the Church (CELAM (Consejo Episcopal Latinoamericano) 1998).[35] According to Gera, who participated in Medellín as a theological expert, this was "a sort of dissociation in the mission of the Church" (Gera 2007, p. 306ff, my translation). Gera also claimed for himself and his fellow supporters of the "Coepalian" view of popular religiosity the inclusion of this issue in Medellín's final document.[36] He had co-authored for the Argentine Episcopal Conference the 1967 National Pastoral Plan, which extolled the theological value of popular religiosity and its "revolutionary" potential for social change, two issues that would be expanded by COEPAL.[37] Büntig sought to remain equidistant between Medellín's twofold division, which he defined as "iconoclastic" and "angelist" (Büntig 1969a, p. 15), but his ideas would be eventually dismissed by the COEPAL theologians as "sociologizing"[38], "Enlightened" (for 18th-century Enlightenment) and foreign-looking (as opposed to rooted in national history and culture).[39] In their view, these were all extremely negative traits, and continue to be regarded to this day as negative traits by TP followers, for whom "Enlightened" (*Iluminista*) is probably the worst insult that they can hurl at an opponent.

The question underlying the ECOISYR *Cuadernos* is to what extent popular Catholicism is an authentic or a doctrinally flawed expression of the evangelical message, and whether the advance of secularization[40]—visibly accelerated in Argentina during the 1960s—would not make popular Catholicism and increasingly "alienating, massifying and social-change impeding" (González [2005] 2010, p. 30, my translation). Büntig's response (Forcat 2016, pp. 5–8; Zanca 2016, pp. 227–29) is framed within the theories of modernization and secularization then in vogue in Europe and the US, theories that he would spearhead in Argentina.[41]

Büntig's hypothesis was that popular Catholicism is "the normal result of the process of institutionalization[42] and inculturation[43], typical of any universal religion that becomes a natural part of a given socio-cultural world [el resultado normal del proceso de institucionalización e inculturación, propios de toda religión universal que se hace parte connatural de un mundo sociocultural determinado]." (Büntig 1969a, p. 17). Everywhere this process unfolds, what he calls "modelled gestures" (empirically observable manifestations of a universal religion: rites, devotions, festivities, cults of the dead, various sacramental practices, etc., such as those analyzed in the survey included in the same *Cuaderno*) "tend to lose their substantive doctrinal strength (they repeat themselves self with ever little value and motivational content) [tienden a perder su vigor sustantivo-doctrinal (se repiten gestos con escaso contenido valorativo y motivacional]" and to popularize. (Idem: 17–18). After this allusion to the Weberian idea of the routinization of the charism (Weber 1946, pp. 262–64), Büntig defines "popular Catholicism" as "all and only those 'patterned' gestures that have been assumed by the Catholic people—at various levels and with various degrees

of identification—as ordinary and spontaneous expressions of their religious experience" (Büntig 1969a, p. 19).

Up to this point, the author seems to lean towards what he has previously defined as the "iconoclasts", but he immediately relativizes his position: those "patterned expressions (...) can be easily—not necessarily—devoid of authentically Christian values and motivations" (Ibid.). However, he highlights those "religious gestures" that are Catholic only "in their external appearance, but which are sustained by non-Christian values (superstitions, religious syncretism, etc.)." On this definition, Büntig based another important distinction, between "real" Catholicism and "doctrinal" Catholicism, or Catholicism adjusted to orthodoxy (Ibid). This conceptualization applies especially to the "inculturated Catholicism [catolicismo inculturado]"[44] of the least modern areas of Latin America—in Argentina, the Northwestern (NOA) and Northeastern (NEA) sections of the country—in which "values and motivations belonging to the pre-Hispanic cultural world" subsist under the disguise of Catholic rituals and devotions (Idem: 21). The clash between these two worldviews is also seen in the popular neighborhoods of the Greater Buenos Aires, where hundreds of thousands of internal migrants from the Northern regions had settled since the 1930s. At the same time, European immigration breathed new life into popular Catholicism through the devotions that they brought from their countries of origin, such as the cult of the Sacred Heart, San Cayetano (the saint patron of jobs), Our Lady of Nueva Pompeya, etc., by now fully incorporated into the national culture (Idem: 27).

This popular Catholicism, both in their Latin American and European streaks, is rooted in the agrarian world. In a society as urbanized and rapidly developing as the Argentine society of the 1960s, those roots tend inevitably to wither and die, and traditional forms, when not straightforwardly rejected by the increasingly unbelieving or just plainly de-Christianized younger generations, tend to be displaced by religious substitutes (horoscopes, divination, "New Age" philosophies, etc.).[45] The retreat of traditional religion, combined with the irruption of neo-pagan religious substitutes, raises some fundamental questions about the value of popular Catholicism. In order to "save" Catholicism from this process of secularization, "that urges us to demagify [sic] the world", Büntig urges his readers to "rescue what is salvageable (...) and progressively suppress what cannot be rescued, (...) in order to eliminate not the true God, but the denaturalization of the sacred Power". (Idem: 70). In other words: Bünting calls to value "doctrinal Catholicism" over "real Catholicism".

This definition brings forth another key distinction, this time between two types of believers: the "naïve" believer of popular Catholicism and the "mature" believer of doctrinal Catholicism, equipped with the right religious education and the "correct motivations".[46] Forming mature believers should be for Büntig the goal of catechesis, and in general, of the Church's pastoral care.

### 3.3. Critique of Büntig's Theses, I: The Sociological Argument (O'Farrell)

Büntig's ideas caused a strong rejection among the COEPAL experts. According to Farrell, the first to react was an old colleague of both of them at ECOISYR, Justino O'Farrell. O'Farrell had begun to define his view of popular Catholicism in a couple of recent articles (O'Farrell 1966, pp. 125–26; 1968), but the most systematic exposition of that view is contained in an unpublished work entitled "Basis for a Research Plan of sociology of religion in Argentina", preserved in Farrell's personal files. Although it is not dated, its classification by Farrell with other ECOISYR documents and the lack of mention in it to the COEPAL allows us to suspect that it must have been written between the end of 1966 and mid-1967, right before the creation of COEPAL.[47] The purpose of this work was to serve as a "basis for discussion" for the "socio-religious research plan" alluded to in the title (O'Farrell n.d., p. 26). It is equally probable that O'Farrell refers here to the plan that would eventually materialize in Büntig's *Cuadernos*, although with a perspective entirely different from what O'Farrell would have wanted. His draft, in effect, is a frontal attack on the theoretical–methodological foundations of religion sociology, as it was understood

by Büntig and his followers. O'Farrell concentrates his attack on two "fundamental facts" that mark "the naiveté and false consciousness in the handling of the sciences of man " in Latin America, and whose overcoming is essential to "get closer to our peoples, who are the real source of social and religious creativity and, therefore, the protagonists of their liberation (O'Farrell n.d., p. 1, my translation) The first of those facts is that the discipline of the sociology of religion, as it was understood in those years, was an "imported cultural product."[48] The second is the inability of this imported product to adequately describe Latin American religious practices (and strictly speaking, any social phenomenon). In other words, the problematic nature "of the process of the relationship between the historical and 'social' processes of our countries and, at the same time, their explanation". (ibid) Despite the Marxist terminology employed (false consciousness, alienation, praxis, etc.), O'Farrell's approach is essentially Hegelian, although in an idiosyncratic way: as antecedents of his "relational dialectic", he also cites Plato, Aristotle, and Jesus Christ (Idem: 6). What O'Farrell takes from Hegel is basically the notion of totality and the rejection of the empiricist conception of knowledge, according to which social facts can be known in themselves, isolated from the totality of historical, political and social relations and processes in which they are inserted.[49]

Of course, this argument exceeds the field of the sociology of religion. O'Farrell's critique of the discipline is part of a broader onslaught that he and his colleagues at *Cátedras Nacionales*[50] were carrying out in those years against "scientist sociology (*sociología cientificista*)" and in favor of the "national sociology" they wanted to establish in its stead in the academe.[51] The main targets of his attack were the theories of modernization, development, and secularization then in vogue in the scholarly world and the inter-American planification and funding institutions for national and regional policies of social and economic development.[52] By treating social facts in isolation, argued O'Farrell, those theories and policies were in fact hiding the historical and political preconditions of Latin American social reality: neocolonialism and dependency. The dependency theory allows O'Farrell to link his methodological criticism to the cultural one: the conceptual schemes that make it impossible to achieve a "total" perception of reality are in fact helping to impede the people's social and political liberation.[53] In his reasoning, there is another, decisive element: O'Farrell's critique was not just directed against the Inter-American capitalist and neo-colonial theories and policies but also Marxist and neo-Marxist theory practices (Idem: 13).[54] This was also a central element of the CN's national-popular ideology. In spite of its leftist-sounding appearance, the anti-Capitalism and anti-Liberal stances that O'Farrell adds to the opposition between national vs. imported culture can be in fact traced back to the counter-revolutionary tradition of integral Catholicism, in which O'Farrell and the other priests of his generation had d been formed.

The main legacy of this tradition can be seen in the conclusions of his essay. Beyond the epistemological and methodological questions, the ultimate foundation of O'Farrell's criticism of the sociology of religion à la Büntig is that it ignores the supernatural dimension of popular religiosity. For this reason, neither popular Catholicism nor the people's liberation can be reduced to the material transformation of social structures: this transformation must be imbued with the transcendent meaning of the Gospel. The People of God, O'Farrell will say, is nothing but the Mystical Body of Christ unfolding in human history (Idem: 17).

### 3.4. Critique of Büntig's Theses, II: The Theological Argument (Gera and Tello)

I have not been able to establish whether O'Farrell's proposal was ever discussed at an ECOISYR meeting. What I can say for certain is that his colleagues at COEPAL's team of theological experts wholeheartedly endorsed it. According to his colleague Farrell, the subject of popular Catholicism was originally brought to the team's attention by O'Farrell, and later elaborated upon by its two leading *periti*, Gera and Tello (Gil and Bender, cit) The issue of popular Catholicism, in its different variants (popular culture, popular piety, etc.) would then become the core of the TPp/TP.

Forcat, whom I follow in this part, has reconstructed the circumstances in which this theological elaboration took place. The decisive moment was at COEPAL's Periti Team meeting of 24 August 1969 (Tello [1969] 2015, p. 39ff; Forcat 2016, p. 8ff). Since Gera could not be present that day, Tello led the discussion, which basically reproduced the main ideas introduced by Gera at COEPAL's Studies Week held at Mallín, Córdoba, in February of the previous year (Gera 2006, pp. 345–99).

The critique of Büntig's thesis can be divided into three parts. First, Tello sets his general frame of reference, which are the notions of faith and Church set forth by *Lumen Gentium* [LG], one of the four dogmatic constitutions of Vatican II. Second, Tello makes a characterization of Latin American popular Catholicism, emphasizing the relationship between faith and popular culture. Lastly, he affirms the priority of faith over the believers' psychological motivations or religious knowledge, thus debunking Büntig's distinction between real and doctrinal Catholicism. To support this point, Tello appeals to a traditional theological argument: the distinction between *credere Deum, credere Deo,* and *credere in Deum,* established by Thomas Aquinas in the 13th century.[55]

In the first part of his argument, Tello describes the supernatural and ecclesial or communitarian nature of the Christian faith:

"Mankind's plan of salvation is a mystery (…) What makes man consciously accept and receive that mystery is faith. (…) By faith, we understand here the theological virtues complex, and even all the moral virtues that depend on them: faith, hope, charity, etc.; (…) faith is what connects, what provides the communion of life with God (…) But that communion of life [with God] simultaneously (…) forms (…) a communion of life with other men (…) Gera insists a lot on this: (…) *Faith within man's heart—that's what the Church really is* (…) Faith, present in man's consciousness, creates the Church, which is the place of mystery, or God's temple. This conception of faith, and above all of the Church, is the most fundamental one, and what illuminates, I think, the whole process". (Tello [1969] 2015, p. 8, my translation).[56]

The second part of Tello's argument is a succinct classification of Latin American Catholicism:

"[There are] *formal* Catholicism, comprised of all the people who have a sense of belonging to the Church; *cultural* Catholicism, which refers to society's Christian values; *political* Catholicism, i.e., the Church as part of the national social body, ingrained in its historical roots; *social* Catholicism, for instance, the importance that baptism and the related issue of the *compadre* and the *comadre* have for the popular classes [and] *implicit* Catholicism, or implicit faith" Ibidem.[57]

Drawing on his ecclesiological perspective (the first part of his argument), Tello can now answer Büntig's question about the theological value of popular Catholicism. Tello's "clear and forceful" response—the third step in his argument—is that *the only criterion for determining the theological validity of any form of Catholicism is the presence of faith*: "If there is a communion with the Mystery—in other words, if there is faith—it is true Catholicism" (Tello [1969] 2015, p. 25). This is why "popular Catholicism is true Catholicism", regardless of "all the cosmological, psychological, primary or secondary motivations Büntig may want to consider" (Ibidem). At the end of the sociological and theological reflection, Tello, Gera and his COEPAL colleagues arrive at exactly the opposite of Büntig's view on popular religiosity.

In this way, Tello lays out the theological foundation of the cultural–historical perspective that O'Farrell had invoked against Büntig's own sociological perspective: "Faith is not identified with its material content (*credere Deum*); faith is above all adherence (*credere Deo*) and tendency (*credere in Deum*) that guides and gives meaning to the life of man. The act of faith does not end in the statement but in God"(Forcat 2016, p. 12, my translation).[58] Faith, moreover, is transmitted in a given culture, in this case, Argentine or Latin American popular culture, which can only be valued through a correct appreciation of that culture's history, "without a hint of sociologism" (Scannone 1995, p. 27, my translation).[59]

"Faith is the affirmation, the movement towards God, towards a salvific Absolute. That seems so clear to me, so absolutely clear in our people! Both in the *criollo* [mestizo native] and the immigrant, the affirmation of life's transcendent meaning, the affirmation that life has a destiny; that life takes place not only in this world but beyond. That life depends on Someone, and that Someone is frequently named with the name "God". It would not be necessary, but the whole sense of God is very explicit. The sense of a God who saves, of a God who in some way communicates, of a God who in some way brings about the fullness of life. It seems to me that all that is built in our people's bones, even if its formulation is flawed. Of course, if they are told about the Plan of Salvation or the Economy of Salvation, they do not understand a word, but that reality is very profound in our people: the reality of a God who fulfills and gives meaning to human life." (Tello [1969] 2015, p. 31, my translation).[60]

The final step in Tello's critique of Büntig's argument links this transcendent sense of popular faith with O'Farrell's critique of secularization theory and cultural colonialism:

"It is said that popular religiosity has a magical sense. This is the grossest mistake; indeed, the magic ones are Büntig and company. That's right. There is an order of the transcendent, of the absolute, of the divine, of the sacred. Magic tries to seize and, by its own means, put at its service that transcendent and that divine. If there is true faith, neither the sacrament, nor the devotion or the holy water are magical, not in the least. Because [popular religious practices] ultimately depend (…) of the action of the Incarnate Word. But the European conception of a rationalized society, which is Büntig conception, says that Man, with his own forces, with the forces of his reason, is capable (…) of dominating nature and (…) fully realize himself. And that leads us to consumer society, to the modern society closed in on itself (Marcuse's magnificent analysis): Man does not need God, Man is self-sufficient for his own fulfillment. So, the process of secularization is exclusion from God. But it is exactly this what is magical: technology is magicalized [sic] Technology is the magic that provides fullness of existence and human life. And so Büntig is the 'magical' one, because he wants to lead Man to a technical life. Popular religiosity is anti-magical insofar as it gives a transcendence that is true transcendence towards the Mystery, the Absolute, the Savior, who is not of this world" (Tello [1969] 2015, p. 40, my translation).[61]

To summarize: Gera-Tello's theological argument complements O'Farrell's methodological critique of Büntig's sociological approach, which analyzes popular Catholicism as a series of gestures or outward expressions of religious experience. The affirmation of the transcendent character of faith incarnated in popular Catholicism—the principle that would later be defined simply as the inculturation of the Gospel in popular piety[62]—and the celebration of its historical and intergenerational transmission through popular culture will be distinctive marks of the TPp and its immediate successor, the theology of culture. Unlike other approaches to popular culture in LT, TPp's rejection of any sociological relativization of popular faith is based on an unquestioned acceptance of the supernatural character of that faith and a deep *Volkisch* sense of popular culture.[63]

This was noted early on by TPp/TP critics. Büntig, in a 1974 article co-signed with his fellow MSTM priest Osvaldo Catena, highlighted what he regarded as the major pitfall of the COPEAL approach to popular faith. His argument is a reprise of the one presented at *Cuadernos* but with an edge of irony, a sign of the souring debates on theological–political issues that had erupted since then among Argentine progressive clerics:

Since this inculturated religion is almost the only thing that simple folks [el pueblo simple] can live and feel, and since all that comes out from the people is good, there is nothing, or very little, to be questioned—one must faithfully respect popular processes. Otherwise, they say, we risk making our people lose the little faith they still have. Those who dare criticize certain sacral and apparently adulterated gestures are Europeanizing [europeizantes] who don't understand our people (Büntig and Catena 1974, p. 107).

It is precisely this uncritical approach to "simple folk" religion that led to the critique of TP as an advancement of cultural and political conservatism over the progressive trends set in motion by Medellín (Ezcurra 1988; González [2005] 2010, pp. 122–23). In any case,

TP's notion of popular piety, which entered papal magisterium with *Evangelii Nuntiandi*[64] and was enshrined in the II CELAM Puebla conference final document of 1979 (where Gera himself played a decisive role as theological *peritus*) became the centerpiece of the Vatican and episcopate's efforts to stem and revert the growth of LT in Latin America.

## 4. Conclusions

The reaction of the COEPAL *periti* against the notion of popular Catholicism defined by the ECOISYR *Cuadernos* was crucial for the articulation of their own vision of popular religiosity, which would in turn be the core of the TPp/TP. In retrospect, it was also the moment when the dividing line between TP and LT was drawn. The fundamental difference between the two lies in the approach to and the concept of popular Catholicism. This approach and concept were articulated for the first time in 1969 at the meetings of the COEPAL's team of theological experts by Lucio Gera, Rafael Tello, and Justino O'Farrell, in the context of a debate over the series of publications on Popular Catholicism coordinated by Aldo Büntig for the ECOISYR. This theological–pastoral debate overlapped with a contemporary theological–political debate on the relation between Socialism and Peronism among Argentina's progressive clerics from 1960–1970 and a broader methodological–political debate on the value of Marxist theory for theological reflection and social change in Latin America. In the curse of the controversies around LT that would shake the life of the Catholic Church in the following two decades, the "Coepalian" conception of popular Catholicism, which emphasizes the supernatural character of popular faith in contraposition to the more sociological, and in many cases, bluntly Marxist assumptions of LT, would gain increasingly greater importance, not only in Argentina and Latin America but also in the papal magisterium, from Paul VI to Francis[65]. These radical differences were observed in his time by critics and supporters of LT and TP, both to bolster or to bring down each of these theological perspectives. Therefore, to use the term "liberationism", clearly inspired by the legacy of LT, to refer to the TPp/PT, distorts the meaning of the latter, which beyond their common historical origin, was historically developed in opposition to the LT. The use of this term is also not justifiable by the common use of the proponents of both schools of the term "Liberation", since for the theologians of the people, the use of "liberation" will have important differentiating notes with respect to its use by the theologians of liberation. This differentiation is summed up in the central reaffirmation of the supernatural quality of popular faith, as expressed in the *sensus fidei fidelium* (Boasso 1974, p. 56) the believers' sense—or supernatural "instinct"—of faith.[66]

**Funding:** This research received no external funding.

**Data Availability Statement:** Not applicable.

**Acknowledgments:** This is in part a translation of "Conceptos analíticos y nativos en el catolicismo posconciliar argentino. El caso de 'liberacionismo', 'Teología de la Pastoral Popular' y 'Teología del Pueblo',' originally presented at the XVIII Jornadas Interescuelas—Departamentos de Historia Universidades Nacionales (the XVIII Jornadas Interescuelas—National Universties History Departments), Santiago del Estero, Argentina, May 10. (forthcoming) This translation was prepared by the author with no external funding. Permission was granted by Coordinación de la carrera Licenciatura en Historia, Universidad Nacional de Santiago del Estero, Argentina (Remeseira 2022).

**Conflicts of Interest:** The author declares no conflict of interest.

## Notes

[1]    In 1985, Juan Luis Segundo, in his response to Cardinal Ratizinger's condemnation of LT for its alleged uncritical embrace of Marxist theory, said that "today many of the most famous theologians in Latin America have nothing than a polite relationship with Marxism" (Segundo 1985, p. 91). By the end of that decade, a shift from an economic-centered analysis to a cultural analysis was also noticeable in LT; see infra, n. 5.

[2]    For the history of COEPAL, see González [2005] 2010. The team of theological experts (*equipo de peritos*) included about half a dozen permanent members, among them Lucio Gera (1924–2012) and Rafael Tello (1917–2002), the two foremost Argentine theologians, sociologists Justino O'Farrell (1924–1981), Fernando Boasso (1921–2015), and Alberto Sily, among others. In addition

to this group, there was a similar number of representatives of the regular clergy and the laity—including some lay and religious women—who changed over the years. The Executive Secretary of COEPAL, Gerardo Farrell (1930–2000), also a sociologist, and his assistant, the future bishop of San Martín, Guillermo Rodríguez Melgarejo, participated in all the meetings.

3    The synchrony–diachrony distinction was borrowed from linguistics (Saussure [1915] 1945, pp. 105–24) and applied to conceptual history by Reinhard Kosleck (Lehmann and Richter 1996, pp. 7–19).

4    Not to be confused with German national–socialism, its Spanish homonym "socialismo nacional" refers to the brand of nationalistic and Government-centered socio-economic and political reforms that were supposed to be implemented in Argentina by a new Peronist administration after the national elections of 11 March 1973. The exact ideological meaning of that term is still a matter of historiographical debate. See Caruso 2022.

5    "Brazilian theology begins to talk about 'people' after we started talking about it, even if they do it in their own way (. . .) [T]he theme 'culture' is our [Argentine TP] contribution; at that time, it made [Gustavo] Gutiérrez really angry" (Azcuy 2006, p. 181, my translation). Gera's reference to Brazilian theology can be understood here as a proxy to LT. The culturalist approach of Gera and his COEPAL's colleagues is the first historical example of what De Schrijver describes as the paradigm shift among Third World liberation theologians from a socio-economic analysis to a cultural analysis; indeed, "culture", understood essentially as *popular* culture), would be the key word of the second iteration of TP, i.e., the theology of culture. This shift "can be gleaned from their different publications, where socio-economic and cultural analysis stand in tensions with each other. In EATWOT [Ecumenical Association of Third World Theologians] conferences since the 1980s, for instance, it has already been made known to the Latin-American liberation theologians that they concern themselves too much with 'economism', i.e., the linking of the praxis of the faith to the struggle against economic oppression. African and especially Asian liberation theologians want to see this method complemented with a revaluation of the local culture. In short, as Latin-American liberation theologians struggle against injustice that oppresses the poor in a capitalistic market-economy, then the theologians of the other 'southern' continents also set out to struggle against injustice that oppresses local cultures by the imposition of uniform rational modern culture." (De Schrijver 1998, p. 3). Despite the instrumental role of non-Latin American theologians in this process, "this shift of attention from the economically oppressed to the culturally oppressed is also seen quite clearly [in] the Latin-American bishops' conferences from 1968 to 1992." (Ibidem) Cfr. (Scannone 1998; Abascal-Jaen 1998). Pope Francis denunciation of "colonialist mentality", i.e., the globalist disregard of peoples' cultural particularities, is an expression of the same line of thought.

6    Regarding the role of Vatican II in the development of Latin American theology, Gera said: "Undoubtedly, it is the Council that practically determines the birth and emergence of theology in all of Latin America, not only in Argentina;". (Azcuy 2014, p. 159, my translation). For a historical survey of Latin American theology since Vatican II, see (Saranyana 2002, chapter 4).

7    In paragraph 48, the 1975 Apostolic Exhortation *Evangelii nuntiandi*—a reflection on the work of the Synod of Bishops convened by the pope the previous year to discuss the issue of evangelization—includes the concept of popular religiosity or popular piety. Paul VI picked up this concept from Pironio's exposition at the Synod (Pironio was then the bishop of Mar del Plata, Argentina) Cfr. (Pironio 1975; Galli 2012a, 2012b). According to Gera, "the fact that the topic 'popular religiosity' came up in the Synod was, I think, Pironio's and the Argentines' [from CELAM] contribution to the Synod" (Azcuy 2014, p. 159, my translation). See also Galli's introduction in (Pironio 2012).

8    Magisterium (Lat. "office of teacher") is the office of authoritatively teaching the Gospel in the name of Jesus Christ (*Dei Verbum* 10). "Those who have the authority to proclaim and teach officially share in the Church's magisterium. Catholics believe that this magisterial authority belongs to the whole college of bishops (as successors to the college of apostolic witnesses) and to individual bishops united with the bishop of Rome" (*Lumen Gentium* 20–25; DV 10). The bishops generally fulfill this magisterium on a day-to-day basis (various kinds of 'ordinary' magisterium). When assembled in an ecumenical council or represented by the pope, they may teach some revealed truth to be held absolutely and definitively (the "extraordinary" magisterium" (O'Collins and Farrugia 2000, p. 148). However, *all* baptized believers have to some degree "a prophetic responsibility for announcing the good news about Christ" because they are "anointed and guided by the Spirit (Jn 14:26, 16:13, Rom. 8:14; 1 Jn 2: 27), *Ibidem.* This is the supernatural foundation of the *sensus fidei* (faith sense or "instinct") of the individual believer and the totality of believers; cfr. (International Theological Commission 2014).

9    The work of Juan Carlos Scannone, S.J. (1931–2019) is a good example of the transformations produced over the past half-century in the use of the term LT and the conceptualization of TP as a current of LT or as an independent theology. Until the late 1990s, the Argentine theologian (whom I am following in this part) described TP (or the equivalent denominations noted above) as a current of LT (Scannone [1982] 1983, 1987a, 1987b, 1998). The use of TL as an "umbrella term" for the different "progressive" theological lines was consistent with the generalized use of it as a synonym for Latin American theology, both inside and outside the Church. However, from his earliest articles on this issue, Scannone also highlighted the differences that existed between TP, in its different diachronic versions, and the most widespread versions of LT (Scannone 1974a, 1975a, 1975b, 1976). The treatment of these differences, but from a philosophical perspective, appears in another facet of Scannone's intellectual work: his contributions to the philosophy of liberation, a kind of secular version of TP (Scannone 1971, 1974b, 1975a, 1990b). At the beginning of the 1990s, accompanying John Paul II's call to the "new evangelization" (cf. the encyclical *Redemptoris missio*) and the spirit of celebration of the fifth centennial anniversary of the evangelization of the Americas, Scannone published one of his most important books, in which the key terms are neither TL nor TP but *culture* and *popular religiosity* (Scannone 1990a). In the 21st century, especially since the election of his fellow Jesuit (Jorge Mario Bergoglio) to the papacy, Scannone increasingly used the term TP, most notably in

the title of his last book (Scannone 2017). From the initial emphasis on LT and the classification of TP as a current of the former, to the emphasis on TP as a distinctive and independent theology, Scannone's intellectual arc also reflects the successive changes of focus of Latin American theology and the magisterium, from the theological–political turmoil of the post-Council era to the Church reforms led by Francis and rooted, according to Scannone and others, in the tenets of TP.

[10] "The book is based on a M.A. thesis [*tesina de licenciatura*] for the Catholic University of Argentina (UCA) School of Theology (in pontifical universities, *licenciatura* is the equivalent to what would be a Master's degree in "civil" universities)". Sebastián Politi, email to and translated by the author, 14 February 2022.

[11] "Sebastián Politi detects a period of particular concentration of forces, of peculiar vitality and creativity (...) on the stage traveled by the Argentine Church between the end of the Second Vatican Council and the period of the military government [1976–1983], which began in 1976. On that stage, he discovers an intense effervescence, revealed, among other phenomena, by the *emergence of a pastoral practice accompanied by a theological reflection that he distinguishes with the denomination 'theology of the people'.*" (Gera in Politi 1992, p. 8, my italics). Since Gera does not mention Segundo's work, he apparently attributes the origin of the term "theology of the people" to Politi.

[12] The full quote reads: "Many agree with Sartre that 'Marxism, as the formal framework of all contemporary philosophical thought, cannot be superseded'." Gutiérrez is quoting Sartre's « Marxisme et philosophie de l'existance » (Geraudy 1961, p. 112).

[13] The concept of *Volkgeist*, and more in general, the philosophical tenets of German Romanticism, appear to have influenced COEPAL theologians mostly through the 19th century Tübingen School of Catholic theology; cfr. (Kasper 2015, pp. 17–18). In the 1950s, Gera studied at Bonn with Arnold Rademacher, who put him in contact with that school, especially the work of Johann Adam Möhler (1796–1831) (Ibidem). That influence was certainly facilitated by the familiarity that any well-educated Argentine of that time would have had with German culture.

[14] "[S]ome would still speak of a theology of liberation, which instead of the Marxist analysis uses a historical-cultural method in which the categories of 'people', 'culture' and 'popular religiosity' acquire salient relevance" (Quarracino 1984a, p. 11, my translation). See also, (Quarracino 1984b, p. 54 ff), where the prelate adds: "but I don't know if these authors would agree to be placed within the so-called 'theology of liberation'." For the documents commented on by Quarracino, see (Congregation for the Doctrine of the Faith 1984, 1986). It is worth mentioning that despite their general anti-scientist and anti-Marxist stances, TP theologians made systematic use of surveys as quantitative research methods.

[15] For a contemporary view of the different currents of the Argentine clergy by one of COEPAL'S leading experts, see (Rodríguez Melgarejo and Gera 1970). "The debate was among progressives; as for the traditionalists, we couldn't really count on them (laughs)". Juan Carlos Scannone, interview with the author, 13 March 2018.

[16] "The debate was among progressives; as for the traditionalists, we couldn't really count on them." Juan Carlos Scannone, interview with the author, 13 March 2018. Besides their political disagreements, the Third World priests were also divided over some important Church issues, above all clerical celibacy.

[17] Although some of these books predated Gutiérrez *Liberation Theology*, it was this work that galvanized the spirit of the times in Latin American progressive Catholicism after the Medellín Conference of 1968 and bolstered the international interest in the subject, which was undoubtedly part of the global attraction to Latin America triggered by the Cuban Revolution. The eclosion of this new brand of Latin American theology was contemporary to the Latin American literary *Boom* of Latin American literature, whose start is usually marked with the publication, in 1967, of Gabriel García Marquez's *One Hundred Years of Solitude*.

[18] Comblin considered that the origins of LT should be traced back to the Christian–Marxist dialogue of the mid-1960s (Comblin 1970, p. 72).

[19] Hosted by the Jesuit-run Institute for Faith and Secularization, El Escorial Encounter, held near Madrid, Spain, between 8–15 July 1972, is considered the international launching of TL. Unlike other encounters, such as the one celebrated the following year in the Spanish city of Toledo (Aldama 1974), El Escorial featured a majority of Latin American theologians. It included presentations by Argentines Aldo Büntig, Enrique Dussel, Juan Carlos Scannone, and the protestant theologian José Míguez Bonino; Brazilians Hugo Assmann and Candido Padin, bishop of Baruru; Chileans Segundo Galilea and Renato Poblete; Peruvians Rolando Ames Cobián and Gustavo Gutiérrez; Uruguayans Héctor Borrat and Juan Luis Segundo; and the Belgian, yet longtime Brazil resident, Joseph Comblin, one of the closest advisors to bishop Hélder Câmara. Among other participants was Chilean Gonzalo Arroyo, organizer of the Christians for Socialism Congress that had taken place in Chile in April 1972. The list of participants also included María Agudelo, Manuel Edwards, Cecilio de Lora, and Noe Zevallos. Büntig, to whom we will return in the last section of this paper, was one of the leaders of the socialist faction of MSTM. Scannone is considered the leading figure of the "second generation" TP (González [2005] 2010, p. 125ss). Interestingly, the Encounter happened during the final years of the Franco dictatorship, which opens a window into the internal dynamics of the Spanish church and the confrontation between Catholic progressives and national Catholics. For an acerbic review of the Encounter from the latter, and a Cold-warrior perspective, see (De la Cierva 1986).

[20] The Jornadas' organizers had originally invited Gutiérrez, but since he was not able to attend, they instead invited Assmann, who was then living in exile in Montevideo. Several decades later, Scannone recalled: "We invited him without knowing much of his thought, so radical, so Marxist" (González and Maddonni 2018, p. 135, n. 79). At the conference, "we discussed whether Marxist analysis was suitable for the moment of Seeing [for the first stage of the see-judge-act method]" Scannone, personal interview, *cit*.

[21]    Comblin, in his analysis of the role of Marxism in Latin American liberation movements, says that during the 1960s the most influential Marxist intellectuals in that regard were heterodox Marxists or Trotskyists, *Monthly Review*-types, such as Paul A. Baran, Paul Sweezy, or Andre Gunder Frank. However, "heterodox leftists such as Franz Fanon were much more influential" (Comblin 1973, p. 117).

[22]    In his historical survey of Argentine sociology of religion, Frigerio places Büntig among the religious sociologists, i.e., confessional sociologists (Frigerio 1993). Although Büntig's goal was to serve the institutional purposes of the Church (a defining characteristic of religious sociology), his theoretical–methodological approach was expeditiously discarded by the local hierarchy in favor of the proposal of the COEPAL theologians. Büntig's true followers are the sociologists of religion whom Frigerio, Soneira, and others identify as forerunners of the scientific autonomy of the discipline from the Church (Frigerio 1993). The trajectories of Büntig and his opponents are equivalent in Argentina to the disputes of religious sociologists at the Conférence Internationale de Sociologie des Religions (CISR), out of which emerged the sociology of religion as a secularized scientific discipline. See *infra*, n. 27.

[23]    Father Gerardo Farrell, executive secretary of both organizations, summarizes the origin and development of the controversy as follows: "[ECOISYR] began the study on popular religiosity, and that effort led to the publication of five volumes [see *infra*, n. 33]. Here a divergence erupts between two lines of interpretation. On the one hand, there is Büntig's approach, very much based on a French sociologist [presumably Pierre-Andre Liégé; see (Forcat 2016, p. 14, n. 66)]. On the other hand, Justino O'Farrell's counterproposal, who regarded Büntig's approach as too secularist." (Gil and Bender 1999, p. 2). With regard to the historical significance of this dispute, the leading historian of TP says: "It was at [ECOISYR] where one of the most important theological-pastoral controversies in Latin American theology would arise: the debate on the theological quality and the pastoral and evangelizing relevance of popular Catholicism" (González [2005] 2010, p. 28).

[24]    "[COEPAL's] new orientation [on the question of popular religion: whether it was superstition or authentic religiosity] was set against Catholic progressives like Büntig. However, it should also be noted that it was Büntig who put the issue [of popular catholicism] on the map". Juan Carlos Scannone, personal interview, cit.

[25]    "Liberation theology emerges in the period spanning since right before Medellín to the immediate post-Medellín, and in the heat of Medellín". (Scannone [1982] 1983, p. 260, my translation).

[26]    "The International Catechetical Study Week which met in Medellín in 1968 (...) was the begin of a process of critically appraising the phenomenon of popular Catholicism" (Prien 2013, p. 223). Prien goes on to praise E.C.O.I.S.Y.R. series *El Catolicismo popular en la Argentina* as one of the first results of that process.

[27]    Those discussions took place within the scope of the Conférence Internationale de Sociologie des Religions (CISR, today Société Internationale de Sociologie des Religions, SISR). Until then, religious sociology had been the paradigm for Catholic sociological research on popular religiosity. Cf. Tschannen, cit.

[28]    Three decades later, Gerardo Farrell would recall: "At that time [the post-council era], the sociological dimension of the Church was very much on the mind of all of us; if the Church had to open up to the world, the human science aimed at the knowledge of society was then a fundamental tool" (Gil and Bender 1999, my translation). At any given time, ECOISYR did not appear to have more than 15 active members. The original group consisted of sociologist priests: Alberto Amato, Aldo Büntig, Justino O'Farrell, Alberto Sily SJ, and Farrell. In February 1967, the group was expanded with about a dozen more names: Santos Benetti, Felipe D'Antona, Osvaldo Musto, Luis Enrique Olivera, Vicente Pellegrini, S.J., Angel Presello, Nicolás Rosato, César Sánchez Aizcorbe, S.J., José María Serra, Alberto Sireau Romain, Gonzalo Becerra, Carlos Lalli, Luis Randisi, and Luis José Gutiérrez.",Gerardo Farrell's personal files, unpublished correspondence. This list is a good synecdoche of the post-conciliar generation: almost half of these priests will join the MSTM, a few will end up leaving the priesthood, and some will have long ecclesiastical careers.

[29]    See, for example, the work of the Catholic Action's Economic and Social Secretariat in the 1930s and 1940s. Statistical surveys were also regularly conducted at the Juventud Obrera Católica (Young Christian Workers movement, JOC).

[30]    In the 1950s, the Catholic weekly *Criterio* and the magazines *Notas de Pastoral Jocista* and *Revista de Teología* regularly published translations and articles by local authors on the subject. This critical mass of articles and practitioners was attractive enough for the young Belgian priest François Houtart, who would go on to become one of the great international references in the discipline, to spend a year of studies in Argentina in 1954 (*Notas de Pastoral Jocista*, 1954), Año VII (Nov.–Dec.): 78–79). During this time, he would produce a number of articles (Houtart 1954, 1955, 1957).

[31]    Soneira has divided the history of socio-religious studies in Latin America into four stages: "(a) the stage of Religious Sociology (1950–1970); (b) a stage where theological-pastoral and/or historical-political reflection of socio-religious studies were predominant (1970–1985); and (c) the stage of the social sciences of religion (1985 onwards). In the first stage, the institutional interests of the churches and an empirical-quantitative methodology at the service of pastoral care were dominant. The second stage was closely linked to the process of change that occurred within the Catholic Church after the Second Vatican Council and its adaptation to the Latin American reality at the General Conference of the Latin American Episcopate held in Medellín. The dependency/liberation axis prevailed as interpretive framework and the historical analysis of the Church/society relationship." (Soneira 1996, p. 112, my translation). The third stage dominant feature is the "academic interest in the study of the religious phenomenon", as opposed to a religious interest (Soneira 2001, pp. 143–44). Frigerio places the heyday of Argentine religious sociology between 1960 and 1980 (Frigerio 1993); in light of the bibliography mentioned above, the chronology should be revised.

32    The Federation of Socio-Religious Research (FERES, for its acronym in French), created in Belgium in 1958, was instrumental in the funding and promotion of new research and publications throughout Latin America in the 1960s. Under its inspiration, Catholic social research institutes sprang up across the region in those years. In Argentina, the most important of them was probably the Jesuit-ran Center for Research and Social Action (CIAS), still in operation. (Zanca 2006).

33    The authors of the other *Cuadernos* were José Severino Croatto and Fernando Boasso (Biblical), Manuel F. Artiles (Psychological), Ciro Lafón and Enrique Dussel (Anthropological), and Dussel and María Mercedes Esandi (Historical). The original project included two more *Cuadernos*—one Theological, to be written by Gera, and another on Pastoral, by several authors—which were never published; that is understandable, and probably inevitable, considering that those were precisely the issues at stake in the Büntig and COEPAL experts' controversy. For an analysis of the series, see (González [2005] 2010, pp. 27–37).

34    The notebook is divided into three parts: "First national survey on popular Catholicism", conducted by students from the Universidad del Salvador School of Political Sciences, presumably Büntig's students; "Motivational interpretation scheme"; and "Essay on functional substitutes for religion in the Federal Capital [Buenos Aires City] and surroundings", plus three statistical annexes.

35    The conclusions of the International Week of Catechesis, held three months before the Medellin Conference in this city, were more openly critical of popular religiosity, which is clearly defined as "conservative, and even partially caused by the dominant superstructures of which the current ecclesiastical organization is part of" (Büntig 1969a, p. 15). Büntig cites these comments as an example of the "iconoclastic" (see *infra*) reaction to popular Catholicism. For the gradual reversal of this judgment in the Latin American magisterium between Medellín and the V CELAM General Conference of Aparecida, Brazil, 2007.

36    "The introduction of the theme 'popular religiosity' in Medellín, at a time when the theology of secularization and Harvey Cox were in full swing, was our own contribution". (Gera in Azcuy 2006, p. 181, my translation). For Gera's role in Medellín, see also (Azcuy 2018).

37    COEPAL was officially created to implement the conclusions of the National Pastoral Plan (Plan Nacional de Pastoral, PNP). The two main theological experts who participated in the elaboration of the PNP, Lucio Gera and Rafael Tello, continued to fulfill their roles as theological experts in the new body.

38    The charge of "sociologizing" (*sociologizante*) and its broader equivalent "scientism" was also launched by the members of the *Cátedras Nacionales* against the mainstream sociology practiced at that time in Argentina's public universities. The *Cátedras Nacionales* intellectuals, led by Justino O'Farrell, supported instead a historical-perspectivist "national sociology". Cfr. *infra*, O'Farrell's argument.

39    "It was rather the European secularist current, the Enlightened modernity, the Enlightenment, Aldo Büntig's writing, that Belgian thing, the French-enlightened thing." (Rodríguez Melgarejo, interview with the author, 2/7/2018).

40    In the writings we are discussing, "secularization" is basically understood in the "classical" sense, as the retreat of established religion in the modern and contemporary world. See Tschannen, cit.

41    Büntig only cites one foreign source (Gustav Mensching), and he criticizes it. However, the concepts he handles show that he was cognizant of the state-of-the-art of sociological and secularization theory. Gera attests to the dissemination of those ideas among Catholics, albeit from a critical standpoint.

42    Cf. the notion of dilemmas of religious institutionalization in the work of the American Catholic sociologist Thomas O'Dea. O'Dea, a disciple of Talcott Parsons, combined concepts from Parsons (deinstitutionalization), Troeltsch (difference between church and sect), and Weber (charisma routinization) in his analysis of the internal organization of the Catholic Church and other religious institutions (O'Dea 1966).

43    Büntig refers here to the process of evangelization of Latin America since the Conquest and to "the survival of many popular forms of our Catholicism, especially in those areas untouched by the flood of immigration.". (Büntig 1969a, p. 20, my translation). Starting in the mid-1970s, the concept of inculturation will become a major topic of theological reflection in the TP tradition and the magisterium. For a survey of the latter, see Martínez Ferrer and Acosta Nassar (2011), and Ballano (2020).

44    See previous note.

45    "In general, the forms that tend to be preserved are those capable of providing a response to the psychological maladjustments and economic insecurity (...) of the modern metropolis" (Idem 27–28).

46    "To avoid these substitutive manifestations, the motivation for personal transformation must be inculcated in people. This leads to a deep commitment, based on the knowledge of the doctrine that is professed." (Idem: 125).

47    It is a 32-page typed manuscript; pages 27–30 are missing and several parts of the text are highlighted with a yellow marker.

48    "[T]he sociology of religion, in its form of activity separate from the concrete social and religious realm to which its material of analysis is inherently attached, is an 'imported product'; we repeat: an 'imported cultural product'." [la sociología de la religión, en su forma de actividad separada del ámbito social y religioso concreto al que su material de análisis está inherentemente unido, es un "producto importado"; repetimos: un 'producto cultural de importación'] ((O'Farrell n.d.): 1, quotation marks in the original).

49    The Hegelian perspective would be fully developed by another member of the National Chairs, Amelia Podetti (Denaday 2013), whose work will play a great influence on the thinking of future Pope Francis. When he was still Archbishop of Buenos Aires, Bergoglio wrote the prologue to Podetti's *Commentary on the Introduction to the Phenomenology of Spirit*. (Podetti 2007).

50    See supra, p. 5.

51    Besides the epistemological approach, the criticism of Büntig's proposal also included strictly methodological questions. "In the overnight polls, I didn't use Büntig method. He used an open question to let people say what they wanted. If you ask that question to simple people, they would answer what they believe you, the pollster, wants to hear. It is like when the teacher is taking an exam: all you want is to pass the exam. In [the sanctuary of] San Cayetano [in Buenos Aires] I learned that it is the other way around—you have to listen to the people, let them talk without asking anything." (Rodriguez Melgarejo, interview by the author, cit.)

52    A few years earlier, O'Farrell had published an article with another pioneer of Argentine religious sociology, Father Antonio Donini. In stark contrast to the ECOISYR article, this one laid out a conventional sociological approach and was theoretically framed in planning theory (O'Farrell and Antonio 1963). The rejection of the economic development paradigm and the theoretical and political radicalization that O'Farrell and other members of his generation experienced in just a few years are indicative of the socio-political ferment of the early 1960s.

53    In Latin America, "the need to consolidate cultural and religious dependency [occurs] through imitation, direct importation and repetition of approaches, problems, and purposes that were originated in countries that are interested in prolonging the system of hegemonic areas and cultures ["la necesidad de consolidar la dependencia de lo cultural y de lo religioso [se produce]a través de la imitación, de la importación directa y de la repetición de enfoques problemas y propósitos originados en el seno de países interesados en prolongar el sistema de áreas y culturas hegemónicas]". (O'Farrell n.d.). The epistemological justification of O'Farrell's religious sociology, as well as of the rest of the so-called national sociology, is rooted in its participation in a collective political project of national liberation. Anticipating (in more than a decade) the poststructuralist bent of postcolonial theory, knowledge, and power are—for O'Farrell and his generation—two sides of the same coin.

54    "[I]n the relationship between society and religion [ . . . ] the adoption of the neo-Marxist and neo-capitalist approach focused on the "underdevelopment" versus "development" dilemma, surreptitiously prolong the conditions of global "alienation"." (O'Farrell n.d., my translation).

55    "In Aquinas' words: 'The act of believing [credere] itself is an act of the intellect assenting to the divine truth at the command of the will that is moved by God through grace." [*Summa Theologica* II-II q.2 a.9, translated by the author]. "Thus, the act of faith is a cognition action. Second, Aquinas distinguishes three aspects of the *one* interior act of faith with the aid of the traditional formula *to believe that God* (*credere Deum*), *to believe God* (*credere Deo*), *to believe in God* (*credere in Deum*) [*ST* II-II q.2 a.2.]. The first part of the formula focuses on what is believed, on the content of faith (. . . ). The second part focuses on the reason for assenting (. . . ). This part of the formula captures the trust aspect of the act of faith. (. . . ). The third part of the formula focuses on the relation of the object of faith to the will—the rational appetite of the believer. The propositions of faith represent the good that believers are lovingly longing for as the fulfillment of their life. What one believes matters heavily for the orientation of one's life as a whole." (Niederbacher 2012, p. 614). By "faith", Aquinas primarily means "the *virtue* of faith, which he defines as "a habit of the mind whereby eternal life is begun in us, making the intellect assent to what is not apparent." [*ST* II-II q.4 a.1.] Faith belongs with hope and charity to the theological virtues. They are called theological virtues because they have God as their object, because they are infused in us all at once by God alone, and because they are made known only by divine revelation in the Holy Scripture. [*ST* I-II q.62 a.1.] (Niederbacher, cit). For Tello's Thomistic conception of faith, see (Bianchi 2012, p. 181ff).

56    Following *Lumen Gentium*, "Tello underlines the temporal and pilgrim dimension of the life of the Church, which *through faith* gives meaning to the path of man in this world" (Forcat 2016, p. 9, italics in the original; my translation). Through the proclamation of the Gospels and the celebration of sacramental life, the historical Church "is making God's plan enter into men" and at the same time is making them "have a *community of life with God.*" (idem: 10, it. in the original, my translation).

57    As Forcat observes, this classification is basically similar to O'Farrell's.

58    Quoting Aquinas, Francis says that in "the act of faith, greater accent is placed on *credere in Deum* than on *credere Deum*" *Evangelii Gaudium* 124).

59    "Our people have existed for centuries, and it is necessary to understand it both in its secular life and in the sense of its wandering. That is why a merely descriptive or sociological method that only captures its current moment is not enough. Nor are the methods created abroad, nor those that take 'modern development' as a parameter" ["Nuestro pueblo tiene una duración de siglos, y es necesario comprenderlo en su vida secular y también en el sentido de su andar. Por eso *no basta un método meramente descriptivo o sociológico que lo capte en su momento actual.* No parecen poder bastar tampoco los métodos creados en el extranjero, ni los que toman como parámetro el 'desarrollo moderno']" (Tello [1969] 2015, p. 31).

60    The original reads: "["La fe es la afirmación, el movimiento hacia Dios, hacia un Absoluto salvífico. ¡A mí eso me parece tan claro, tan absolutamente claro en el hombre nuestro! Ya sea en el criollo o en el inmigrante: la *afirmación del sentido trascendente de la vida*; que la vida tiene un destino; que la vida se juega no solamente acá sino más allá. Que la vida depende de Alguien, y eso también se nombra frecuentísimamente con el nombre 'Dios'. No haría falta, pero es muy explícito todo el sentido de Dios. El sentido de un Dios que salva, de un Dios que de algún modo comunica, de un Dios que de algún modo realiza la plenitud de la vida. *Eso me*

*parece que está metido hasta los tuétanos en nuestra gente*, aunque la formulación sea muy defectuosa. Por supuesto, si se les habla de Plan de Salvación, o de Economía de Salvación no entienden nada, pero la realidad es así muy honda en la gente nuestra: de un Dios que plenifica y da sentido a la vida del hombre"]".

61　The original reads: " Se afirma que la religiosidad popular es de sentido mágico. *Ese es el error más craso*; porque precisamente los mágicos son Büntig y compañía. Exactamente. Hay un orden de lo trascendente, de lo absoluto, de lo divino, de lo sacro. La magia pretende con sus propias fuerzas apoderarse y poner a su servicio ese trascendente y ese divino. Si hay verdadera fe, ni el sacramento, ni la devoción, ni el agua bendita, ni todo eso es mágico. Porque no es sino en función (…) de la acción del Verbo encarnado, en último término. En cambio en la concepción europea de sociedad racionalizada, en la cual se mueve Büntig, se dice esto: el hombre con sus propias fuerzas, con las fuerzas de su razón, es capaz de (…) dominar la naturaleza (…) y realizarse plenamente. Y eso los lleva a la sociedad del consumo, a la sociedad moderna cerrada sobre sí misma, (todo el magnífico análisis de Marcuse): el hombre no necesita de Dios, el hombre se basta a sí mismo para su propia realización. Entonces el proceso de secularización es exclusión de Dios. Pero precisamente eso es lo mágico. La técnica se magiciza [sic]. La técnica es lo mágico que le da la plenitud de existencia y de vida humana. Entonces el mágico es Büntig cuando quiere llevar al hombre a una vida técnica. Y la religiosidad popular es anti-mágica en cuanto da una trascendencia que es verdadera trascendencia hacia el Misterio, el Absoluto, el Salvador, que no es de este mundo"]".

62　See supra, n. 14. For the critique of TP's notion of "people" from a TL perspective, see (Segundo 1975, pp. 207–32). See also (González [2005] 2010, pp. 113–16). For a historical survey of popular religiosity, popular Catholicism, and popular piety in Latin America, see (Prien 2013).

63　See supra, n. 13.

64　See supra, pp. 5–6 and n. 7.

65　For the summary of Francis' view on popular faith, see his programmatic apostolic exhortation *Evangelii Gaudium*, the evangelizing power of popular piety, paragraphs 122–126.

66　See supra, n. 8, in fine.

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
