# Peer review of "Analytical and Native Concepts in Argentina’s Post-Conciliar Catholicism: The Case of “Liberationism”, “Popular Pastoral Theology”, and “Theology of the People”"

_religions, doi:10.3390/rel13111110_

Round 1

Reviewer 1 Report

no comments

Author Response

Thank you for your consideration

Reviewer 2 Report

The subject of the article is very good and relevant for the studies of contextual theologies, especially in liberation theology. It is very helpful that the author decides to compare liberation theology and theology of people developed in Argentina. The authors' presentation of theology of people is well articulated, especially its discussion about the popular Catholicism. But I think that authors's presentation of liberation theology is limited to one author and view. In page 18, the conclusion of the author that the main different between LT and TP is related to their view of popular Catholicism. He/she argues that this difference is "fundamentally irreconcilable". This seems a natural conclusion, considering the authors' references of LT, quoting mainly G. Gutiérrez. However, there is a lack of other great names of LT that can't be dismissed because they also develop the concept of popular Catholicism, such as Clodovis Boff and José Comblin. I recommend that the author of this text also offers an explanation of popular Catholicism of LT as he/she did explaining this concept for the TP. If LT has a fundamentally different view of popular Catholicism, the author must show evidence with quotes from several liberation theologians on this concept. I just mentioned two who deal with it and there are more. 

Sentences in Spanish after being presented in an English translation are not helpful and totally unnecessary. This creates confusion and makes the text longer and hard to read. If there is a word that can't be well translated into English, it must be used in the original with a footnote explaining why. This is enough. In addition, there are some parts, especially in the footnotes, that there are full sentences in Spanish that need translation.  

Overall, the article is good and can be create contribution to the field after some adjustments. 

Author Response

Thank you for your thoughtful remarks; they helped me hugely to see the weak points of my argument and to try to do my best to correct them.

 I think that authors’ presentation of liberation theology is limited to one author and view. In page 18, the conclusion of the author that the main different between LT and TP is related to their view of popular Catholicism. He/she argues that this difference is "fundamentally irreconcilable". This seems a natural conclusion, considering the authors' references of LT, quoting mainly G. Gutiérrez. However, there is a lack of other great names of LT that can't be dismissed because they also develop the concept of popular Catholicism, such as Clodovis Boff and José Comblin. I recommend that the author of this text also offers an explanation of popular Catholicism of LT as he/she did explain this concept for the TP. If LT has a fundamentally different view of popular Catholicism, the author must show evidence with quotes from several liberation theologians on this concept. I just mentioned two who deal with it and there are more. 

My original formulation was indeed too extreme, and this observation helped me to refocus. My point is not popular Catholicism per se, which as Reviewer#2 indicates, was also addressed by LT authors. My purpose in this article is not to survey the peculiarities of the periti’s conception of popular Catholicism vis a vis those of other LT theologians, but to delve into a contextual analysis of their debate with Büntig in order to expose the mainstays of their theological thought. In the terms of my theoretical framework, my focus is on the diachronic analysis of TP and LT at the time of the disputes described in the paper. Further developments of LT, less closely connected with Marxist theory, as well of the so-called convergence between TP and LT, are of course relevant from a synchronic perspective, but do not invalidate the diachronic analysis of their differences—and similarities--within the linguistic context of their historical emergence.

Sentences in Spanish after being presented in an English translation are not helpful and totally unnecessary. This creates confusion and makes the text longer and hard to read. If there is a word that can't be well translated into English, it must be used in the original with a footnote explaining why. This is enough. In addition, there are some parts, especially in the footnotes, that there are full sentences in Spanish that need translation.  

I eliminated the Spanish original version from the body of the article and confined it to a few instances in the notes that I think could be helpful for readers who want to check my own translation with the original.

Reviewer 3 Report

The outline of the article is clear and coherent. One learns a lot from the anaysis of the the debate on the theological schools and thr development of the concept of liberation.

To mark the difference between LT and TpP/TP is a worthwhile enterprise and the line of argument contains important details and insights. The stated distinction is well done. 

To mark the distinctive feature of LT mainly its anti-modernism an dproximity to Marxism as Loewy does seems to undifferenciated for different reasons. First modernity cannot be reduced to capitalism and second liberation theology (understood as as umbrella term) shows assorted characteristics that  - at least in its history of reception. To make this point so strongly at the beginning is misleading.

"The affirmation of the transcendent character of popular Catholicism and of the value of its historical transmission through popular culture" (p. 18) is a very important "signum" of TP and merit of the article. Still a critical point lies in the assumption that popular religion per se is pureley transcendental and has the immediate sense ("instinct" of the truth. What about so many hybrid forms of Christian faith one finds in popular religion as well as in secular form of spirituality? The concept of the "sensus fidelium" still lacks a clear explanation and is added very quickly in the end (p. 19) without unfolding it deeper.

I can not follow the conclusion in the end that states the "irreconcilable difference" (p. 18) as I find that there exist bridging positions inbetween, that politics and faith are both necessary parts of liberation, and that theology and sociology are two ways to approach a scientific question.

But all this is more a matter of theological points of view (with a post-secular and maybe as well one-sided view of European theology) to be discussed than a matter concerning the quality of the contribution which merit is to create awareness of the distinctive aspects and the specific character of TP. 

Author Response

Thank you for your thoughtful remarks; they helped me hugely to see the weak points in my argument and to try to do my best to correct them. 

To mark the distinctive feature of LT mainly its anti-modernism and proximity to Marxism as Loewy does seems to undifferenciated for different reasons. First modernity cannot be reduced to capitalism and second liberation theology (understood as an umbrella term) shows assorted characteristics that  - at least in its history of reception. To make this point so strongly at the beginning is misleading.

This is a valid argument, and it overlaps with the remarks on my characterization of LT made by Reviewer#1. However, this is Löwy’s definition of LT, and it is that definition on what the analytical term “liberationism” normally used by historians and sociologist of religion is based. My goal is not to discuss the validity of Löwy’s definition vis a vis other definitions of LT, other different authors, but to discuss how Löwy’s definition is used in the historiographical and sociological literature through the use of the term “liberationism” (and that is why it must appear at the beginning of the article.

The affirmation of the transcendent character of popular Catholicism and of the value of its historical transmission through popular culture" (p. 18) is a very important "signum" of TP and merit of the article. Still a critical point lies in the assumption that popular religion per se is pureley transcendental and has the immediate sense ("instinct" of the truth. What about so many hybrid forms of Christian faith one finds in popular religion as well as in secular form of spirituality?

This is of course a valid critique, but I believe it’s a critique of the periti’s conception of popular religion. In this paper I am not concerned with the validity of that conception but how that conception, or more specifically,  the way in which the periti reached that conception, provides a valid explanation to differences between TP and TL (again, basically as defined by Löwy)

The concept of the "sensus fidelium" still lacks a clear explanation and is added very quickly in the end (p. 19) without unfolding it deeper.

I expanded the information on this concept in the notes.

I can not follow the conclusion in the end that states the "irreconcilable difference" (p. 18) as I find that there exist bridging positions in between, that politics and faith are both necessary parts of liberation, and that theology and sociology are two ways to approach a scientific question.

See my first response to R#1

Round 2

Reviewer 2 Report

The author made major revisions in the text, considering my previous analysis.